# Unravelling the Complexity of Colorectal Cancer: Heterogeneity, Clonal Evolution, and Clinical Implications

**DOI:** 10.3390/cancers15164020

**Published:** 2023-08-08

**Authors:** Nadia Saoudi González, Francesc Salvà, Javier Ros, Iosune Baraibar, Marta Rodríguez-Castells, Ariadna García, Adriana Alcaráz, Sharela Vega, Sergio Bueno, Josep Tabernero, Elena Elez

**Affiliations:** 1Vall d’Hebron Institute of Oncology, 08035 Barcelona, Spain; nsaoudi@vhio.net (N.S.G.);; 2Oncology Department, Vall d’Hebron Hospital, 08035 Barcelona, Spain

**Keywords:** metastatic colorectal cancer, tumor heterogeneity, biomarkers, clonal evolution, ctDNA

## Abstract

**Simple Summary:**

Metastatic colorectal cancer is a complex, prevalent, and life-threatening disease influenced by various factors that affect its progression, evolution, and treatment responses. Tumor heterogeneity, stemming from genetic and non-genetic factors, impacts tumor development and therapy effectiveness. This feature can be assessed by computational analysis of next-generation sequencing to understand spatial tumor evolution and diversity. Analyzing circulating tumor DNA allows the study of temporal heterogeneity by real-time monitoring of tumor changes and treatment response. Different models explain the origins of this heterogeneity, emphasizing complex molecular pathways. This review examines these concepts and focuses on the clinical implications of clonal evolution and tumor heterogeneity.

**Abstract:**

Colorectal cancer (CRC) is a global health concern and a leading cause of death worldwide. The disease’s course and response to treatment are significantly influenced by its heterogeneity, both within a single lesion and between primary and metastatic sites. Biomarkers, such as mutations in *KRAS*, *NRAS*, and *BRAF*, provide valuable guidance for treatment decisions in patients with metastatic CRC. While high concordance exists between mutational status in primary and metastatic lesions, some heterogeneity may be present. Circulating tumor DNA (ctDNA) analysis has proven invaluable in identifying genetic heterogeneity and predicting prognosis in *RAS*-mutated metastatic CRC patients. Tumor heterogeneity can arise from genetic and non-genetic factors, affecting tumor development and response to therapy. To comprehend and address clonal evolution and intratumoral heterogeneity, comprehensive genomic studies employing techniques such as next-generation sequencing and computational analysis are essential. Liquid biopsy, notably through analysis of ctDNA, enables real-time clonal evolution and treatment response monitoring. However, challenges remain in standardizing procedures and accurately characterizing tumor subpopulations. Various models elucidate the origin of CRC heterogeneity, highlighting the intricate molecular pathways involved. This review focuses on intrapatient cancer heterogeneity and genetic clonal evolution in metastatic CRC, with an emphasis on clinical applications.

## 1. Introduction

Colorectal cancer (CRC) is a global health problem, being the second most common cause of death worldwide and causing almost 1 million deaths per year [1]. For decades, several international cooperative efforts have been initiated to better understand the underlying basis driving this disease. From the earliest studies that helped to understand the evolution from adenoma to adenocarcinoma, in which *APC*, *TP53*, and *KRAS* mutations were recognized as driving and clonal mutations, to the most modern single-cell sequencing studies, the importance of inter- and intratumoral heterogeneity has been highlighted as a key factor in the natural history of CRC [2,3].

CRC displays significant diversity regarding histological classification and can be categorized into three primary histological subtypes: adenocarcinoma, mucinous adenocarcinoma, and signet ring cell carcinoma. Among these, adenocarcinomas are the most prevalent tumors in the colorectal region. However, the other two subtypes are less common and exhibit unique features. Patients with mucinous adenocarcinoma and signet ring cell carcinoma often experience an earlier age of onset, present at more advanced stages, and have a higher likelihood of lymph node and peritoneal metastases [4,5].

The biomarkers established to guide treatment in patients with first- and second-line metastatic CRC (mCRC) are marked by mutations in the MAPK pathway, such as mutations in *KRAS*, *NRAS* exon 2, 3, and 4, and *BRAF*, in addition to the study of deficient mismatch repair (dMMR)/microsatellite instability (MSI) [6,7]. When choosing a biological treatment for patients with mCRC, the heterogeneity of these molecular alterations at the intratumoral level may have significant effects. As a result, the choice of sample (tissue or blood) and location (primary tumor, circulating tumor DNA [ctDNA], or metastasis) to be studied may have an impact on their treatment. Today, it is widely accepted that CRC is a heterogeneous and molecularly complex disease, and that knowledge of its molecular characteristics has profound implications for the management of CRC patients to improve clinical outcomes and offer better personalized therapy.

Cancer is not a single disease, but several. Each patient’s case is unique, and each tumor cell is unique and distinctive. The concept of tumor heterogeneity is a broad one and can cover several aspects. Tumor heterogeneity can be genetic, referring to differences in DNA sequences and gene expression, or non-genetic, referring to differences in the functional and metabolic characteristics of tumor cells brought about by various factors such as epigenetic regulation or post-translational modification. Both types of heterogeneity can affect tumor development, therapy response, and disease outcome. Moreover, different levels of heterogeneity can be defined. While spatial heterogeneity refers to differences that may exist within a single patient affected with metastatic cancer, such as differences between a primary tumor and metastatic lesions, or between various genetic subpopulations within a tumor lesion, interpatient heterogeneity refers to variations seen across tumors of the same histological type in different patients [8]. The dynamic nature of mCRC as it develops over time and gives rise to diverse subclones as a result of fluctuating selective pressures brought on by a cancer’s lifetime of therapy is highlighted by temporal heterogeneity.

Cancer heterogeneity is one of the greatest challenges for mCRC research. High genetic diversity within a tumor can give rise to subclones with selective advantages, leading to the development of drug-resistant and metastatic cancer. This intratumoral heterogeneity is associated with tumor relapse and, in moving towards optimal cancer treatment and personalized oncology care, it is essential to comprehend subclonal variety, clonal evolution, and intrapatient cancer heterogeneity. This review provides an overview of intrapatient genomic mCRC heterogeneity and clonal evolution, examines the clinical implications, and presents current methods for addressing these challenges [9,10].

## 2. Assessing Tumoral Heterogeneity and Clonal Evolution

It is well established that even if two patients have the same tumor with the same histology, they will respond differently to treatment and have distinct oncological histories. This is particularly true for CRC patients due to various tumoral genesis processes, some of which may be partially mediated by the patient’s unique exposome [11]; interpatient heterogeneity underlies this concept. In this review, we focus instead on intrapatient heterogeneity. As scientific understanding has grown, it is now apparent that cancers are not a collection of identical cells but rather tumor cells that differ from one another both genetically and epigenetically. These distinct populations of cancer cells are known as clones. In addition, the tumor microenvironment (TME, primarily constituted by cancer-associated fibroblasts, vascular cells, and immune cells) actively interacts with cancer cells. This interaction directly or indirectly controls cell proliferation, angiogenesis, nutrient and oxygen delivery, as well as mechanisms of immunomodulation and immune evasion [12]. This close interaction with the TME exerts selective pressure on tumor clones, thereby modulating clonal evolution. Tumor heterogeneity is strongly fueled by this clonal evolution, which we define as variations in the allele frequency of various cell populations over time, caused by mutations and natural selection by environmental selective pressures, which might include both the TME and cancer treatments. Ultimately, taken together, this alters the clonal selection and evolution of tumor cells, which in turn affects the heterogeneity of cancer cells.

The question arises of how can we study genetic heterogeneity at the level of the primary tumor? Classically, a genomic study is performed at different spatial points of the primary tumor, from different points of the metastatic lesions and, by means of sequencing techniques (currently next-generation sequencing [NGS] techniques), with computational and mathematical studies to infer the evolution of the accumulation of genetic alterations [13]. This allows us to go back in time to very beginning of a cancer. We are able to generate a phylogenetic tree showing the trunk or most ancestral genetic alterations in time, present in all tumor cells, or clonal or private mutations, and subsequently subclonal or private alterations present in a specific group of cells. This genetic diversity is the motor of tumor genetic heterogeneity. Thus, we understand tumor genetic heterogeneity to be caused by the presence of different subclones in the same tumor that share the same genomic characteristics. These studies at the tissue level allow us to take a snapshot and understand the process that has occurred in the past to arrive at the tumor we have in the present [13]. The accuracy of this method depends on thorough representative sampling of both the primary tumor and metastases. DNA sequencing can be used to track the evolutionary steps of metastasis and gain insights into the genetic and molecular mechanisms that contribute to cancer dissemination. Studies that contain numerous metastases from the same patient have shown high inter-metastatic heterogeneity [14,15,16,17]. The substantial amounts of tissue required for studying the relationships between cancer cells, the microenvironment, and the host can be obtained from research autopsies. Nonetheless, a recent review compiling multiple studies on tumor heterogeneity and clonal evolution in cancer based on autopsy studies showed an under-representation of CRC [18].

The concept that cancer develops through clonal evolution was first put on the table in 1976 in an article by Peter Nowell [19]. Cancer is a clonal evolution process, driven by repeated rounds of selection by TME and treatments, and produces tumors with various genetic and molecular alterations. This process is highly dynamic. High-throughput technologies have made it possible to profile specific cancer genomes for actionable genomic alterations, and these determinations can be made without the need for single or multiple biopsies thanks to the study of ctDNA. The percentage of circulating DNA fragments that are directly derived from tumor cells and can be separated from the total amount of circulating free DNA in blood is known as ctDNA. ctDNA is detected in blood at higher levels than other circulating biomarkers and is linked to better sensitivity and specificity for the detection of cancer [20,21]. In rough terms, we can classify ctDNA detection techniques into two groups: tumor-informed and tumor-agnostic (also known as tumor-uninformed or plasma-only ctDNA) [22]. As the name implies, tumor-informed testing uses information about specific mutations in a patient’s tumor to build probes for evaluating the patient’s plasma. Traditionally, ctDNA methods employing only plasma were thought to be less sensitive. Recent innovations in plasma-only testing have increased their sensitivity by incorporating other markers such as methylation or epigenomics [23]. Liquid biopsy studies enable us to recognize the presence of various clones in real-time and to forecast future responses to targeted therapy, just as tissue studies enable us to take a picture of all the genetic modifications that have accumulated and in what order. Also, we must keep in mind that acquiring a tissue sample for a genetic study biases the results because it is not feasible to take multiple samples from the same patient’s multiple metastatic lesions due to the invasiveness of the same process. Thus, we can obtain a more accurate image of all the clones present in a patient using ctDNA, which gives a broader insight than a single tissue biopsy. The population of clonal cells in patients with mCRC is also not always the same since, as previously mentioned, both TME and treatment place a strong selection pressure on the tumor. In addition, we have to take into account how the sample we are studying for heterogeneity has been obtained. The main differences in cancer heterogeneity between a biopsy and a surgical report for colon cancer are related to sample size and representation. Biopsies offer a smaller and limited sample, potentially overlooking various tumor subclones, while surgical reports analyze a larger tumor specimen, offering a more comprehensive understanding of tumor heterogeneity.

The use of ctDNA allows us to track this clonal evolution secondary to the treatments in real-time to improve precision medicine. In fact, in a translational study of patients with HER2-amplified mCRC who participated in the HERACLES clinical trial (trastuzumab plus lapatinib), the Italian group studied ctDNA at different points in the trial in 30 patients. The researchers performed genetic and functional investigations after comparing the ctDNA with radiological findings, tumor biopsy findings, and autopsy findings (from one patient). Using samples and cell models from eight metastases in a patient chosen for a post-mortem investigation, they discovered that pharmacological vulnerabilities and evolutionary trees could be correlated in ctDNA, and that there was a relationship between lesion size and the number of different metastases that contribute to plasma ctDNA [24]. 

One of the major limitations of studying clonal evolution via ctDNA is that ctDNA plasma shedding is not detected in all patients with mCRC, as not all tumors release tumor DNA into the blood. The location of metastatic lesions is known to impact the amount of ctDNA shed. mCRC patients who have liver metastases exhibit higher detection rates compared to those with peritoneal or lung metastases [25].

The generally accepted method for determining spatial intratumoral heterogeneity involves whole-exome sequencing or high-coverage multigene panels run on several biopsy sites. However, for a precise evaluation of intratumoral heterogeneity, ultra-deep sequencing might be necessary [26]. More focused strategies could be used to enhance patient management in addition to -omic studies intended to characterize CRC in greater detail. Thanks to their lower cost and quicker turnaround times, NGS-focused panels have recently been suggested for application in clinical practice [27,28]. 

It is challenging to compare diverse research approaches and to homogenize data due to the lack of standardized procedures for the investigation of intratumoral heterogeneity [29]. Accurate characterization of tumor cell subpopulations remains problematic. Mutant-allele tumor heterogeneity, studied as mentioned above, is commonly used as a measure of genomic intratumoral heterogeneity. A mutant-allele tumor heterogeneity score is defined as the ratio of the median absolute deviation to the distribution’s median given a distribution of variable allele frequencies [30]. However, to perform an accurate study of intratumoral heterogeneity, we need to describe the percentage of cancer cells in the sample (also known as the tumor purity), the formation and growth of cancer subclones, as well as the proportion of each non-cancerous cell type in the tumor microenvironment. Multi-omic data integration of genomic, epigenetic, transcriptomic, and proteomic information is required to fully comprehend this complexity [31]. Also, adequate computational methods to model the data and deduce the underlying biological states of a tumor are needed [31]. A cutting-edge method for assessing intratumoral heterogeneity is deconvoluting bulk-tumor gene expression profiles using single-cell RNA sequencing (scRNA-seq) [32]. The bulk analysis obtained by NGS informs us about alterations shared by many cells, giving insights into the history of the tumor. scRNA-seq can detect unique, recent, and active events, exponentially increasing the ability to correctly understand intratumoral heterogeneity. For example, a recent study evaluating scRNA-seq of 198 cancer cell lines from 22 cancer types identified 12 recurrently heterogeneous expression programs related to diverse biological processes, characterized the landscape of heterogeneity within various cancer cell lines, and identified recurring patterns of heterogeneity shared between tumors [33]. 

## 3. The Genesis of Tumor Heterogeneity 

This section covers various colon cancer carcinogenesis models, based on reliable evidence. The most informative way to view them is as complementary models rather than competing ones, and that they most likely coexist at various stages of the disease. It is also crucial to keep in mind that the different models are founded on various methods of research and problem-solving, ranging from genomic methods (Big Bang) to scRNA-seq techniques.

### 3.1. Three Major Molecular Pathways of Colorectal Cancer 

To understand CRC heterogeneity and clonal evolution, it would be interesting to first review CRC genesis. The challenging nature of this disease is widely accepted, with its multi-stage development that involves genetic and epigenetic alterations in the cells lining the colon and rectum. About 70% of CRC cases are initiated with an adenomatous polyp as a precursor of the disease. Yet, depending on the molecular pathways at play, the progression of such polyps into invasive cancer differs, as CRC can be driven through at least three primary molecular mechanisms. This process typically begins with hyperproliferative epithelium, where there is an increase in the number of cells in the colon and rectum, followed by the development of focally dysplastic crypts, where there are areas of abnormal cell growth and organization. 

The next stage is the appearance of visible growths, known as macroscopically evident tubular adenomas, which can progress to become increasingly dysplastic and/or villous adenomas [34]. In some cases, these adenomas can develop into invasive cancer. According to the multistep model of carcinogenesis proposed by Vogelstein et al. in 1988, the development of CRC involves the successive accumulation of genetic changes, such as somatic mutations and chromosomal abnormalities and particularly chromosomal instability (CIN) [2]. *TP53*, *APC*, *KRAS*, *PIK3CA*, *BRAF*, and *SMAD4* are examples of specific oncogenes and/or tumor suppressor genes that are frequently mutated in tumors with CIN [35]. About 15% of CRCs are caused by mutations in DNA repair genes involved in the mismatch repair system (dMMR). 

Widespread hypermethylation in gene promoter areas (also known as the CpG island methylator phenotype, or CIMP) is the third pathway to CRC. Both of the latter two pathways cause significant instability in simple repeating nucleotide sequences, known as microsatellite instability (MSI). There are two categories of CIMP+ tumors: CIMP-high tumors, which are linked to *BRAF* mutations and *MLH1* promoter methylation, and CIMP-low tumors, which are linked to *KRAS* mutations [36].

It is worth mentioning again that not all the origin of heterogeneity is due to genetic alterations. A recent study investigated intratumor heterogeneity in CRC and showed that genetic and epigenetic variations, along with transcriptional plasticity, contribute to intratumor heterogeneity [37]. However, the authors suggest that intratumor genetic ancestry only minimally impacts gene expression traits and subclonal evolution. Most intratumor gene expression variation is not strongly heritable but rather “plastic”, indicating widespread transcriptional flexibility within CRC. Somatic expression quantitative trait loci analysis identifies some genetic controls of expression, mainly clonal within a tumor, but a considerable proportion of CRCs show no evidence of subclonal selection, suggesting that most genetic intratumor variation does not have significant phenotypic consequences.

### 3.2. The Importance of the Tumor Microenvironment

CRC represents a finely tuned example of how the development of tumors depends on the interaction of tumor cells with their TME, in addition to the existence of numerous critical genomic alterations. The interaction of every element of the TME, including the various immune system cellular subtypes, the vascular endothelial cells, cancer-associated fibroblasts, and nerve cells, as well as the inclusion of the microbiota in this role-play, undoubtedly contribute to the selective pressure that leads to the emergence of distinct subclones within the tumor and is therefore implicated in the evolution of CRC tumors [12,38]. Therefore, rather than being only the result of genetic alterations, the development of CRC is a complicated interaction between genetic changes and the TME.

The identification of four CRC subtypes (CMS1–CMS4) according to the creation of a consensus molecular subtype (CMS) classification is noteworthy. Based on bulk transcriptome signatures, the International Consortium published the CMS, which categorized CRC as CMS1 (MSI immune), CMS2 (canonical), CMS3 (metabolic), and CMS4 (mesenchymal). CMS1 tumors are highly immunogenic, whereas CMS4 cancers have a poor prognosis because of their mesenchymal stroma and potent TGF-signaling patterns [39]. As cancer-associated fibroblasts are created, dysregulated TGF signaling is a key factor in their ability to develop tumors and evade the immune system. The CMS classification relies on data obtained through bulk sequencing, which by nature lacks the resolution to investigate CRC tumors and their complex microenvironment at the cellular level required to uncover molecular fingerprints in small but important cell populations. Several bulk expression investigations have shown that stromal cells can mask important signals from other significant cellular phenotypes within the CRC spectrum, which can have an impact on CRC classifications, revealing the challenges posed by the fact that gene expression profiles of bulk-tumor tissue samples represent the sum of signals from the cancer cells and tumor microenvironment [40,41]. The CMS example is an important snapshot of the implication of MSD and intratumoral functional heterogeneity in the current understanding of the molecular biology of mCRC.

Recently, a study in nasopharyngeal carcinoma postulated that cancer cells in nasopharyngeal carcinoma patients exhibit a diverse range of subtypes that interact and compete in a genetically and phenotypically diverse ecosystem, contributing to malignant progression and clinical implications [42]. It highlights the importance of understanding cancer evolution and its impact on patient outcomes. The study emphasizes that cancer should be viewed as a multidimensional spatiotemporal “ecological and evolutionary unity” rather than just a genetic disease, and suggests that normal–polyp(adenoma)–different stages of colon cancer can be considered as an ecological disease [42].

### 3.3. The Cancer Stem Cell Model

According to the cancer stem cell hypothesis, only a small proportion of a tumor’s cells are able to initiate and drive tumor growth. The bulk of any particular neoplasm is made up of non-tumorigenic cells that are incapable of metastatic seeding or tumor growth; however, the cancer stem cells are the fundamental building block of tumor growth and metastatic spread [43]. Several studies have shown that a fraction of LGR5+ stem-like tumor cells are responsible for the progression of CRC [44,45,46]. The leucine-rich repeat-containing G-protein coupled receptor (LGR5) is a protein that enhances the canonical Wnt/β-catenin signaling pathway and is present in normal stem cells in various tissues including the intestine. This receptor is considered a marker of stem cells due to its potential to generate self-organizing crypt/villus structures in vitro called organoids or “mini-guts” from single isolated LGR5+ cells [47]. In experimental settings, a landmark study found that LGR5 cancer stem cells are unnecessary for primary CRC growth but required for metastasis formation [45]. Subsequently, it was postulated that metastases are started by disseminated differentiated tumor cells that, once reaching the liver, through plasticity, produce LGR5+ cancer stem cells [48].

Recent research may alter how this stem cell hypothesis has been understood so far. A transcriptomic analysis using scRNA-seq of the genes linked to poor prognosis in CRC patients showed that the majority of these genes accumulate in a particular cell type known as high-relapse cells (HRC), which have an associated gene signature enriched in genes related to cell migration and cell-to-cell adhesion, and that differentiated from LGR5+ stem cells [49]. The authors developed mouse models of CRC with metastatic recurrence following primary tumor surgery, discovering that HRCs remained “hidden” after primary tumor excision, giving rise to various cell types, including LGR5+ stem-like cells, and causing overt metastatic disease. They identified and selectively removed this cell population using Emp1, a marker gene for HRCs (encoding epithelial membrane protein 1). Mice were disease-free following surgery thanks to genetic ablation of EMP1-high cells, which prevented metastatic recurrence. The authors concluded that cell-state dynamics of residual disease in CRC and anticipated that therapies targeting HRCs may help to avoid metastatic relapse.

### 3.4. The Big Bang Model

Genomic instability, a trait of mCRC as outlined above, is predicted to result in substantial intratumoral heterogeneity. Though not all clones will live, those that are particularly “fit” may be able to outcompete the rest. It is possible that the only differences between subclones are small, non-selective molecular changes. Therefore, differentiating between “driver mutations”—which promote tumor genesis—and “passenger mutations”—which have no impact on the fitness of the clone—is important [50]. Therefore, it is conceivable that there must be an interaction between the different clone populations.

In 2015, a sequencing study performed by Sottoriva et al. in hundreds of tumoral glands to map the spatial distribution of genetic alteration revealed that at the initial clonal evolution stage the evolution is neutral. This means that multiple subclones can coexist without any one clone being fitter than the others. The clonal richness and diversity observed in mCRC at the primary tumor level can be attributed to this neutral initial evolution, which is referred to as the “Big Bang” theory [51].

The Big Bang model of CRC growth proposes that the mutations that cause tumor development and progression occur early in CRC and that the timing of a mutation is the primary factor in determining its frequency, rather than selection. In this model, all major clones persist during growth, and metastatic dissemination begins early during tumor development, reflecting intratumoral heterogeneity. This theory has been supported by other studies, including in silico studies [26,52,53,54].

It should be noted that evolution that is effectively neutral (or nearly neutral) does not imply that selection is absent. Instead, subclones with a fitness benefit can be uncommon or emerge late without enough time to increase to a detectable frequency. The totality of these findings points to the possibility of fluctuating evolutionary rates, with periods of fast mutation accumulation followed by relative stasis [55].

### 3.5. Beyond Darwin’s Evolutionary Theory

Darwin established the theory in 1859 that explains how inheritance, selection, and genetic variability contribute to species variety, evolution, and extinction [56]. Over a century later, Nowell pondered if Darwin’s theory could account for the processes of carcinogenesis, cancer response, and adaptability to treatment after observing the variability in cancer [19].

Although it has been demonstrated in multiple studies that gene-centered Darwinian principles can explain the evolutionary trajectories of tumors, recent research suggests that other evolutionary theories are required to account for the complete range of evolutionary behaviors in cancer [57,58]. For example, there is growing evidence that cancer exhibits macroevolutionary leaps that are likely broken up by periods of microevolutionary gradualism, as described with the Big Bang Theory [51]. A wider range of evolutionary models must also be considered due to evidence of inconsistent inheritance patterns among cells, as well as the impact of neutral evolution, cellular plasticity, and the TME on cancer [58].

## 4. Clinical Application of Clonal Evolution and Tumor Heterogeneity in mCRC

In this section, we review the biological basis of clonal evolution and heterogeneity in CRC and assess the clinical applicability in mCRC and impact on day-to-day clinical decisions.

### 4.1. Correlation of Biomarkers between the Primary Lesion, Metastases, and ctDNA

The choice of first-line therapy in patients with newly diagnosed mCRC depends significantly on the results of biomarker testing [7,59]. Testing for *KRAS*, *NRAS* exon 2, 3, and 4, and *BRAF* V600E mutations as part of the work-up for confirmed CRC in the metastatic setting is recommended by the National Comprehensive Cancer Network (NCCN) guidelines, the European Society for Medical Oncology (ESMO), and the Pan-Asian adapted ESMO consensus guidelines, and all patients with CRC (including those with localized status) are tested for MSI [7,59,60,61,62]. Some guidelines consider testing other biomarkers such as *HER2* amplification after at least first-line progression. Other oncogenic drivers including *ALK* and *ROS1* gene fusions, *PIK3CA* mutations, and *HER2*-activating mutations are not (yet) recommended outside of clinical trials. Although the incidence of *NTRK* fusions in mCRC is extremely low (0.5%), testing is still advised when feasible [6,61].

Through the paired analysis of surgical specimens of primary and metastatic lesions, numerous studies have investigated the consistency between the mutational status of key pathway genes, such as *RAS*, *BRAF*, and *PIK3CA*, in paired mCRC tissue lesions. Most investigations have focused on liver metastases, showing that mutation patterns were highly concordant (>90%) independently of the temporality of the metastases (metachronous or synchronous) [63,64,65,66,67]. Despite this, some authors report a high frequency of heterogeneity in *KRAS* mutations in synchronous primary metastasis in the liver [68]. Studies have found a strong correlation between primary tumors and peritoneal, brain, and ovarian metastases for driver genes [69,70,71]. Discordance rates may correlate with the location of metastases, with a higher discordance risk for lymph node metastases [72]. Some authors emphasize the significance of taking the degree of necrosis of a sample into account, while examining lymph nodes as a possible explanation for this discordance [71].

As previously described in this review, the use of ctDNA facilitates investigation of temporal and spatial genetic heterogeneity in mCRC. Moreover, the quantification of ctDNA through the analysis of the mutant-allele fraction (MAF) has been demonstrated to be an independent prognostic factor in *RAS*-mutated mCRC patients [73]. Several studies have performed correlation analyses between the results of driver mutation analysis in solid biopsy samples and the results in ctDNA. One study examined the association between the concordance rate of ctDNA results and tissue biopsy results and the tumor burden in 221 patients analyzed with OncoBEAM (a ctDNA test that uses the BEAMing digital PCR technology testing *RAS*). Patients had metastatic disease in the liver (n = 151), peritoneum (n = 25), or lung (n = 45), with concordance rates of 91%, 88%, and 64%, respectively. They observed concordance rates ≥ 90% in patients with liver metastases alone and peritoneal or lung metastases alone with a baseline longest diameter ≥ 20 mm and/or with ≥10 lesions [25]. In a second study, researchers assessed 72 consecutive patients with mCRC using the IdyllaTM Biocartis test (a fully automated real-time PCR-based platform testing *KRAS*, *NRAS*, and *BRAF*) and found an overall agreement of 82% between ctDNA and standard tissue-based NGS analyses. Concordance was higher in treatment-naïve patients (86%) and in the subgroup with liver metastases (96%) [74]. In a third study, the researchers evaluated the feasibility of a blood-based ctDNA assay for detecting *KRAS* mutations in patients with mCRC (using the OncoBEAM assay based on BEAMing digital PCR able to analyze mutations in *KRAS* and NRAS). They found that the concordance rate between ctDNA and tissue biopsy results was 89%, indicating that ctDNA analysis can be used to guide the selection of appropriate therapy [75]. In summary, studies show a high correlation, although it should be borne in mind that this approach has limited application for patients with mCRC without detectable ctDNA in their blood due to less shedding of tumor DNA. Detection rates also vary based on the location of metastases, with liver metastases showing higher rates of detection than peritoneal or lung metastases.

### 4.2. Clonal Evolution in Molecularly Selected mCRC Patients

Clonal evolution and the appearance or re-emergence of clones resistant to targeted therapy reflect a two-sided coin. On the one hand, the mutations that are responsible for this clonal evolution may be mechanisms of resistance and explain disease progression; however, they may also represent a vulnerable target for an additional line of targeted therapy. In this section, we assess several clinical examples in mCRC patients, as summarized in Table 1.

#### 4.2.1. BRAF Mutant mCRC Patients

The *BRAF*-V600E mutation occurs in 8–12% of mCRC patients, and leads to constitutive activation of the kinase, resulting in poor response to chemotherapy and short overall survival [87,88,89]. Anti-VEGF therapy combined with chemotherapy has shown improved clinical outcomes in *BRAF*-V600E mCRC patients, while the addition of an anti-EGFR to a chemotherapy backbone is not recommended [7,90,91,92]. In contrast to the excellent outcomes in melanoma, single-agent *BRAF* inhibitors showed little to no clinical activity in patients with *BRAF*-V600E-mutated mCRC [86,93].

In an attempt to understand the differences in response to these drugs as monotherapy between different histologies, preclinical studies revealed a resistance mechanism involving feedback activation of EGFR in the presence of *BRAF* inhibitors [94,95]. This has led to testing of various combinations of *BRAF* and MEK inhibitors, anti-EGFR agents, and chemotherapy in clinical trials with *BRAF*-mutant mCRC patients [86,96,97,98]. However, most combinations have shown limited clinical activity, with modest improvement in overall response rates (ORR) and median progression-free survival.

The pivotal BEACON trial demonstrated that combining *BRAF* inhibition with anti-EGFR therapy was more effective than irinotecan-based chemotherapy in treating *BRAF*-V600E mCRC in the refractory setting. The trial compared encorafenib–cetuximab with or without the MEK inhibitor binimetinib (doublet or triplet therapy) with irinotecan–cetuximab-based chemotherapy. Median overall survival for both the triplet and the doublet was 9.3 months compared to 5.9 months for the control arm [84]. The MEK inhibitor’s role in this population remains to be defined.

Studies have found that after *BRAF* inhibitor combinations acquired resistance mechanisms—such as *MAP2K1*, *GNAS*, *ARAF*, *PTEN*, *ERBB2*, *MEK*, and *EGFR* mutations, as well as *KRAS*, *MET*, and *EGFR* amplification—have been detected in both plasma and tissue samples from patients [96]. As an example, ctDNA analysis data from the BEACON trial have recently been reported. The most common acquired resistance alterations were *KRAS* and *NRAS* mutations, as well as *MET* amplification, observed in both doublet and triplet treatment arms. In comparison, the control arm showed much lower rates of these alterations (0% in the control arm compared to 40% in the target therapy arm). The acquired alterations detected at the end of treatment seem to be subclonal, and a significant number of patients who developed these mutations exhibited multiple alterations simultaneously [99]. These data confirm the importance of selective environmental pressure with targeted treatment.

Analysis of ctDNA is a useful tool for identifying those alterations and thus could be useful for tracking and developing targeted therapies to overcome resistance to *BRAF* inhibitors. As an example, *MET* amplification has been reported as a druggable acquired resistance mechanism after a *BRAF* inhibitor combination, and switching to a MET inhibitor plus *BRAF* inhibitor improved outcomes in some cases with confirmed *MET* amplification [100,101,102]. Figure 1 illustrates this therapeutic possibility after detection of resistant clones in ctDNA. These cases highlight the need for diagnostic tools to track molecular alterations and guide treatment choices in the dynamic evolution of *BRAF*-V600 mutant tumors. Analysis of ctDNA could also help identify patients who may benefit from a BRAF inhibitor rechallenge [100,103].

#### 4.2.2. Clonal Evolution in KRASG12C Mutant mCRC Patients

The *RAS* gene family, including *HRAS*, *KRAS*, and *NRAS*, plays a crucial role in cell signaling pathways driving growth, differentiation, and survival. Mutations in *KRAS*, particularly in exon 2 codons 12 and 13, are found in 40% of mCRC cases and are associated with a worse prognosis [104,105]. These mutations also serve as negative predictive biomarkers for response to EGFR inhibitors, such as cetuximab and panitumumab [106,107,108]. Despite efforts for many years, *RAS* mutations have proven challenging, remaining undruggable until recently. However, there has been progress in transforming *RAS* mutations from being negative predictive factors for EGFR inhibitors response into positive predictive biomarkers for targeted therapies.

The *KRAS*-G12C mutant cysteine is situated near a pocket found in the inactive GDP-bound state of KRAS. Sotorasib (AMG 510), a first-in-class target therapy, exhibits a unique mechanism by selectively and permanently inhibiting the kinase activity by trapping it in the inactive GDP-bound isoform. Sotorasib was tested in an all-solid tumor phase 1 clinical trial [76], with 42 patients with *KRAS*G12C mCRC included. The response rate in mCRC with monotherapy was 7%, compared to other tumor types, such as non-small cell lung cancer (NSCLC), with response rates of up to 32%. When translational studies were performed to assess these differences between histologies, EGFR signaling reactivation was found to be behind the dominant mechanism of resistance in *KRAS*-G12C inhibitors in mCRC models, compared with NSCLC [109]. Based on this evidence, clinical trial designs with *KRAS*G12C inhibitors subsequently included EGFR inhibition. Early data presented at ESMO 2022 from the CodeBreaK 101 phase 1b trial exploring the combination of sotorasib and panitumumab in chemo–refractory *KRAS*G12C-mutated mCRC showed promising antitumor activity in the fully enrolled dose expansion cohort of 40 patients with refractory mCRC, demonstrating a confirmed ORR of 30% [77]. Other *KRAS*G12C inhibitors that have been developed, such as adagrasib, demonstrated clinical activity in combination with cetuximab in heavily pre-treated patients with mCRC, with an ORR of 46% and a median response duration of 7.6 months [110].

The history and the mechanisms involved in the reactivation resistance of the EGFR pathway resemble those observed in the development of *BRAF* pathway inhibition. The KRYSTAL-10 phase 3 study is ongoing, comparing adagrasib–cetuximab with standard chemotherapy regimens in the second-line setting, while other trials explore *KRAS*G12C inhibitors in combination with different agents targeting *KRAS*-G12C.

#### 4.2.3. Clonal Evolution in RAS/BRAF Wild-Type CRC and the Importance of Anti-EGFR Therapy Precision Selection

In clinical practice for mCRC, it is essential to identify the status of *RAS* mutations as they serve as negative predictive biomarkers for response to EGFR inhibitors [106,107,108]. Multiple clinical trials involving cetuximab and panitumumab have demonstrated that not only mutations in *KRAS* exon 2 but also mutations in exons 3 and 4, as well as exons 2, 3, and 4 of *NRAS* (expanded *RAS* analysis), are associated with a lack of response to anti-EGFR monoclonal antibodies [106,108]. As a result, the European Medicines Agency restricts the use of panitumumab and cetuximab to patients with *RAS* wild-type mCRC. Moreover, recent data from the PARADIGM trial demonstrated longer overall survival in patients with *RAS* wild-type and left-sided mCRC treated with first-line mFOLFOX6 plus panitumumab compared to bevacizumab [111]. Negative hyper-selection using ctDNA gene alterations, such as *KRAS*, *NRAS*, *BRAFV600E*, *PTEN*, *EGFR*, *HER2*, *MET*, and gene fusions, may help identify patients who would benefit from panitumumab over bevacizumab regardless of primary tumor sidedness, suggesting its potential usefulness as a biomarker for anti-EGFR therapy resistance [112]. These data suggest the importance of clones of resistance detected by ctDNA and the relevance of clonal heterogeneity in clinical practice.

In first-line treatment of *RAS* wild-type mCRC, combining chemotherapy with anti-EGFR monoclonal antibodies is a standard approach, especially in left-sided mCRC [7,59]. Analyses of ctDNA have shown that *KRAS* mutant clones, initially present at subclonal levels, increase under anti-EGFR therapy [113]. It is hypothesized that anti-EGFR therapy reduces the population of sensitive cells, allowing resistant clones to become predominant until disease progression. Subsequent non-EGFR-based treatments in the second-line may restore the sensitive clones to some extent, creating the foundation for anti-EGFR rechallenge [113]. Recent molecular evidence emphasizes the intratumoral heterogeneity of CRC and the dynamic clonal evolution driven by treatment pressure. The emergence of *RAS* mutations during disease progression on first-line chemotherapy plus anti-EGFR monoclonal antibodies can be followed by a decrease in the MAF of these mutations. The decay of *RAS* and other resistant clones in ctDNA during non-EGFR-based treatment has been calculated to have a half-life of between 3.7 and 4.7 months. This decay of *KRAS* mutations in the blood after discontinuing anti-EGFR antibody treatment indicates clonal evolution during therapy. This time has been used in the past to empirically test EGFR inhibitors’ rechallenge, with low ORRs seen with this method [114]. Figure 2 schematically illustrates the clonal evolution during rechallenge with EGFR inhibitors.

Following advances in ctDNA analysis, this technique was brought into the rechallenge setting. The initial data are from a study with retrospective analysis of ctDNA (CRICKET), and a subsequent study used these data prospectively to select patients who could benefit from this strategy (CHRONOS). The multi-center phase 2 CRICKET trial evaluated a rechallenge strategy using cetuximab and irinotecan in patients with wild-type *RAS* and *BRAF* mCRC who had acquired resistance to first-line irinotecan- and cetuximab-based therapy [115]. Out of 28 enrolled patients, the ORR was 21% with six partial responses and nine disease stabilizations. The retrospective analysis of baseline ctDNA showed that the presence of *RAS* mutations correlated with shorter progression-free survival, suggesting that using ctDNA can aid in patient selection for this rechallenge approach. Prospective studies were thus performed, and in the CHRONOS phase 2 trial patients with tissue *RAS* wild-type tumors who had previously received anti-EGFR therapy underwent ctDNA-based screening, resulting in the exclusion of 31% of patients with resistance mutations in ctDNA [116]. Among the enrolled patients, 63% achieved disease control, demonstrating the potential of using ctDNA-guided anti-EGFR rechallenge with anti-EGFR as a safe and effective approach for refractory mCRC patients. Larger randomized trials are needed to further compare this strategy with standard-of-care therapies.

## 5. Conclusions

Understanding clonal evolution and intrapatient tumor heterogeneity is crucial for optimizing treatment strategies in mCRC patients. From a practical, day-to-day practice perspective, the key biomarkers needed in clinical practice need to have a high correlation rate between primary and metastatic lesions, particularly liver metastases. However, some differences in data of tissue from different sites exist, depending on the metastatic site, or the uniformity in all subclones, the size of biopsies, and the inherent risk of sampling error. This heterogeneity (or technique limitation) has recently been shown to be manageable thanks to ctDNA studies and has implications for clinical outcomes of targeted therapies, as recently reported with the PARADIGM trial ctDNA sub-analysis in the first-line setting of mCRC patients treated with EGFR inhibitors. Analysis of ctDNA also allows for the investigation of temporal genetic heterogeneity in mCRC and for tracking clonal evolution. Understanding clonal evolution is critical for the accurate treatment of molecularly selected mCRC patients such as with *BRAF*-V600E and *KRAS*G12C mutant tumors, as the detection of the emergence of resistant clonal populations is crucial for adapting target treatment. Combination therapies targeting multiple pathways have shown promising results in overcoming some resistance mechanisms, such as the recurrent reactivation of the EGFR pathway observed when *KRAS*G12C or *BRAF*V600 is inhibited. Monitoring the emergence and decay of resistant clones through ctDNA analysis has implications for treatment strategies and rechallenge with EGFR inhibitors.

Overall, an improved understanding of clonal evolution and tumor heterogeneity provides valuable insights into the molecular complexity of mCRC. Integrating genomic sequencing, ctDNA analysis, and scRNA-seq techniques into routine clinical practice is crucial for a comprehensive understanding of tumor heterogeneity. The use of novel technologies, such as spatial high-plex imaging multi-omics, will help us to deepen our understanding of intrapatient and intratumor heterogeneity, especially from a spatial point of view [117]. Clinical trials need to focus on understanding and manipulating cancer evolution by considering the process of clonal evolution and competition between different subsets of drug-sensitive and resistant clones. By incorporating next-generation sequencing to track clonal dynamics and adopting novel treatment strategies such as intermittent dosing or cycling different treatments, clinical trial designs can be improved to exploit and benefit from clonal evolution for better patient outcomes [118]. Further research in intrapatient heterogeneity and clonal evolution is warranted to unlock the full potential of personalized medicine in mCRC.

## Figures and Tables

**Figure 1 cancers-15-04020-f001:**
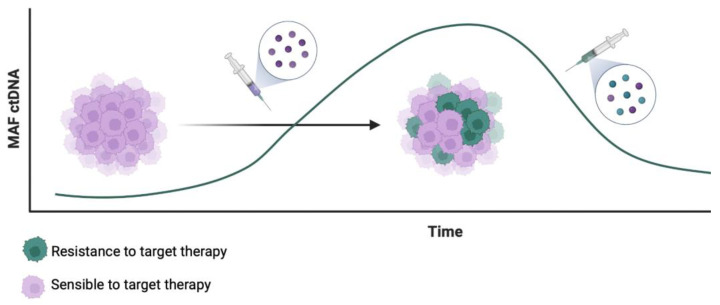
Therapeutic possibilities after detection of resistant clones in ctDNA in the targeted treatment of mCRC.

**Figure 2 cancers-15-04020-f002:**
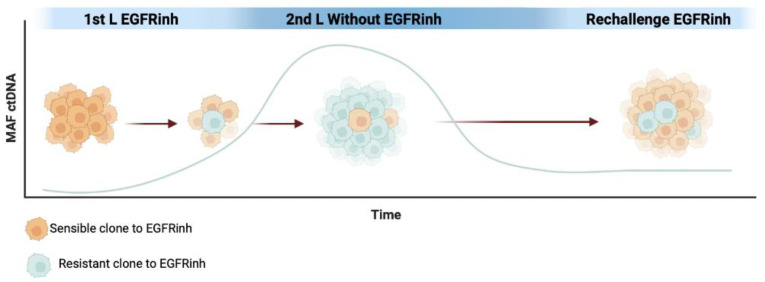
MAF representation of EGFR inhibitor (EGFRinh) resistance mutations in ctDNA per clonal evolution during the first line, and later with the release of the pressure of selective EGFR treatment.

**Table 1 cancers-15-04020-t001:** Completed clinical trials for molecularly selected metastatic CRC patients with *KRAS*G12C or *BRAF*V600E mutations.

Drug(s)	Trial Name	Phase	Number of Pts	Efficacy Outcomes
** *KRAS* ** **G12C**				
Sotorasib (AMG 510) [76]	NCT03600883 CodeBreaK100	1	42	ORR: 7.1% (3/42)/mPFS: 4 months
Sotorasib + panitumumab [77]	NCT04185883 CodeBreaK101	1/2	40	ORR: 30%/mPFS: 5.7 months
Adagrasib(MRTX849) [78]	NCT03785249 KRYSTAL-1	2	43	ORR: 19%/mPFS: 5.6 months
Adagrasib + cetuximab [78]		2	28	ORR: 46%/mPFS: 6.9 months
JNJ 74699157 [79]	NCT04006301	1	4	Stopped due to skeletal muscle toxicities and lack of efficacy
LY3499446	NCT04165031	1/2	5	Early termination due to unexpected toxicity
GDC-6036 [80]	NCT04449874	1	43	ORR: 20%
JAB-21822 [81]	NCT05009329	1/2	9	In all population: ORR: 50%
** *BRAFV600* **				
Encorafenib–binimetinib–cetuximab (1stL) [82]	NCT03693170 ANCHOR	1/2	95	ORR: 47.4%/mPFS: 5.8 months
Encorafenib–cetuximab–alpelisib [83]	NCT01719380	2	52	ORR: 27%/mPFS: 5.4 months
Encorafenib–cetuximab–binimetinib [84]	NCT02928224 BEACON	3	224	ORR: 26.8%/mPFS: 4.5 months
Encorafenib–cetuximab [84]		3	220	ORR: 19.5%/mPFS: 4.3 months
Dabrafenib–trametinib [85]	NCT01072175	1/2	43	ORR: 12%/mPFS: 3.5 months
Vemurafenib–cetuximab [86]	NCT01524978	2	27	ORR: 2%/mPFS: 3.7 months

ORR, overall response rate; mPFS, median progression-free survival.

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
