# Peer review of "Unravelling the Complexity of Colorectal Cancer: Heterogeneity, Clonal Evolution, and Clinical Implications"

_cancers, 2023, doi:10.3390/cancers15164020_

Round 1
Reviewer 1 Report
GOOD JOB!WHEN WE DISCUSS ABOUT ETEREOGENITY WE SHOULD MENTION THE MOST COMMON SITUATIONS,THE PATHOLOGY WHO CLASSIFICATION AND STAGING.FURTERMORE A REFER TO THE HISTOLOGY DIFFERENCES BETWEEN BIOPSY AND SURGICAL REPORTS COULD BE "A POINT IN FAVOR"
Author Response
Dear Dr. Catherine Shao,
We are sincerely grateful for the chance to submit a revised draft of our manuscript. The insightful comments provided by the reviewers have been immensely helpful, and we have diligently incorporated their suggestions into the revised version of the manuscript. All the changes made in response to the reviewers' comments are appropriately highlighted throughout the document.
Please find attached a point-by-point response addressing the reviewers' comments and concerns.
Comments from Reviewer 1
Comment 1: WHEN WE DISCUSS ABOUT ETEREOGENITY WE SHOULD MENTION THE MOST COMMON SITUATIONS,THE PATHOLOGY WHO CLASSIFICATION AND STAGING.
Response: Thank you for pointing this out. We agree with this comment. Therefore, we have added a description of this topic in the introduction section, page 2, line 46.
Comment 2: FURTERMORE A REFER TO THE HISTOLOGY DIFFERENCES BETWEEN BIOPSY AND SURGICAL REPORTS COULD BE "A POINT IN FAVOR"
Response: We agree with this comment. Therefore, we have added a description of this topic in the introduction section, page 4, line 157.
Reviewer 2 Report
As the authors indicate, this review focuses on intrapatient cancer heterogeneity and genetic clonal evolution in metastatic CRC, with an emphasis on clinical applications.
Several points should be noted as below.
1) “Cancer evolution ” and “cancer heterogeneity” have been a hot topic of research in recent years. Maybe there is no unified definition for it yet, one thing we can confirm, that is, the meaning of “cancer evolution ” is not equal to “cancer clonal evolution”, and “cancer heterogeneity” also is not the same to “cancer genetic heterogeneity”. It should be clarified.
2) The authors wrote that “Cancer is not a single disease” , “... both TME and treatment place a strong selection pressure on the tumour.”, etc. A recent paper proposes that the nature of cancer is not a genetic disease but an ecological disease: a multidimensional spatiotemporal "unity of ecology and evolution" pathological ecosystem (https://www.thno.org/v13p1607.htm). In this paper, the author also proposes that normal-polyp(adenoma)-different stages of colon cancer can be considered as an ecological disease. Therefore, there is a question that I would like to discuss here. Such relevant viewpoints might further enhance our understanding of cancer evolution and TME.
3) One recent report showed that “ most genetic intratumour variation in CRC has no major phenotypic consequence and that transcriptional plasticity is, instead, widespread within a tumour.”, it should also be discussed.
4) As to “clonal evolution in the progression of CRC”, can the authors provide a pattern diagram?
5) What is point of “The cancer stem cell model” without mentioning “evolution” in this section . In fact, CSCs is regarded as an evolutionary process.
6) The “Big Bang Model”, I think this ignores the fact that cancer is an evolutionary and ecological process.
Author Response
Dear Dr. Catherine Shao,
We are sincerely grateful for the chance to submit a revised draft of our manuscript. The insightful comments provided by the reviewers have been immensely helpful, and we have diligently incorporated their suggestions into the revised version of the manuscript. All the changes made in response to the reviewers' comments are appropriately highlighted throughout the document.
Please find attached a point-by-point response addressing the reviewers' comments and concerns.
Comments from Reviewer 2
Comment 1: “Cancer evolution ” and “cancer heterogeneity” have been a hot topic of research in recent years. Maybe there is no unified definition for it yet, one thing we can confirm, that is, the meaning of “cancer evolution ” is not equal to “cancer clonal evolution”, and “cancer heterogeneity” also is not the same to “cancer genetic heterogeneity”. It should be clarified.
Response: Thank you for your valuable comment, we appreciate your insight into the terminology surrounding these topics. We agree that there might not be a universally agreed-upon definition yet. We have distinguish between "cancer heterogeneity" and "cancer genetic heterogeneity" to avoid any confusion in page 2 line 66.
Comment 2: The authors wrote that “Cancer is not a single disease” , “... both TME and treatment place a strong selection pressure on the tumour.”, etc. A recent paper proposes that the nature of cancer is not a genetic disease but an ecological disease: a multidimensional spatiotemporal "unity of ecology and evolution" pathological ecosystem (https://www.thno.org/v13p1607.htm). In this paper, the author also proposes that normal-polyp(adenoma)-different stages of colon cancer can be considered as an ecological disease. Therefore, there is a question that I would like to discuss here. Such relevant viewpoints might further enhance our understanding of cancer evolution and TME.
Response: Nice article! Thank you for the suggestion. We added this information in page 6, line 288.
Comment 3: One recent report showed that “most genetic intratumour variation in CRC has no major phenotypic consequence and that transcriptional plasticity is, instead, widespread within a tumour.”, it should also be discussed.
Response: What an interesting study, thanks for the reference. We have summarised it on the page 5 line 248.
Comment 4: As to “clonal evolution in the progression of CRC”, can the authors provide a pattern diagram?
Response: Thanks for the suggestion, just note that figure 1A and 1B already explain this concept.
Comment 5: What is point of “The cancer stem cell model” without mentioning “evolution” in this section. In fact, CSCs is regarded as an evolutionary process.
Response: Thank you for the point, in this section we wanted to discuss the recent recognition of HRCs and the relevance to traditional CSC theory.
Comment 6: The “Big Bang Model”, I think this ignores the fact that cancer is an evolutionary and ecological process.
Response: You have raised an important point here. However, we believe that the “Big Bang theory” of evolution has been validated and exposed by experiments in a pre-clinical manner, whereas to date (and to the best of our knowledge, apologies if this is not the case) there is no pre-clinical evidence of the "ecology" theory in colon cancer.
Round 2
Reviewer 2 Report
The authors have fully answered my concerns.